# Vitamin C in Home Parenteral Nutrition: A Need for Monitoring

**DOI:** 10.3390/nu12061667

**Published:** 2020-06-03

**Authors:** Julienne Grillot, Sabrina Ait, Charlotte Bergoin, Thomas Couronne, Emilie Blond, Catherine Peraldi, Didier Barnoud, Cécile Chambrier, Madeleine Lauverjat

**Affiliations:** 1Nutrition Intensive Care Unit, Hospices Civil de Lyon, CEDEX, F-69495 Pierre Benite, France; khedidja-sabrina.ait-abderrahim@chu-lyon.fr (S.A.); charlotte.bergoin@chu-lyon.fr (C.B.); didier.barnoud@chu-lyon.fr (D.B.); cecile.chambrier@chu-lyon.fr (C.C.); 2CarMeN Laboratory, INSERM, INRA, INSA Lyon, Université Claude Bernard Lyon 1, CEDEX, 69921 Oullins, France; emilie.blond@chu-lyon.fr; 3Department of Gastroenterology, Hospices Civils de Lyon, CEDEX, F-69495 Pierre Benite, France; thomas.couronne@chu-lyon.fr; 4Biochemistry Department, Lyon Sud Hospital, Hospices Civils de Lyon, F-69495 Pierre Benite, France; 5Centre Agréé de Nutrition Parentérale à Domicile, Hospices Civils de Lyon, CEDEX, F-69495 Pierre Benite, France; catherine.peraldi@chu-lyon.fr (C.P.); madeleine.lauverjat@chu-lyon.fr (M.L.)

**Keywords:** vitamin C, home parenteral nutrition, commercial multivitamin preparation

## Abstract

To date, there are no recommendations about screening plasma vitamin C concentration and adjust its supplementation in patients on long-term home parenteral nutrition (HPN). The aim of this study was to evaluate vitamin C status and determine if a commercial multivitamin preparation (CMVP) containing 125 mg of vitamin C is sufficient in stable patients on HPN. All clinically stable patients receiving HPN or an intravenous fluid infusion at least two times per week for at least 6 months, hospitalized for nutritional assessment, were retrospectively included, for a total of 186 patients. We found that 29% of the patients had vitamin C insufficiency (i.e., <25 µmol/L). In univariate analysis, C-reactive protein (CRP) (*p* = 0.002) and intake of only 125 mg of vitamin C (*p* = 0.001) were negatively associated with vitamin C levels, and duration of follow-up in our referral center (*p* = 0.009) was positively associated with vitamin C levels. In multivariate analysis, only CRP (*p* = 0.001) and intake of 125 mg of vitamin C (*p* < 0.0001) were independently associated with low plasma vitamin C concentration. Patients receiving only CMVP with a low plasma vitamin C level significantly received personal compounded HPN (*p* = 0.008) and presented an inflammatory syndrome (*p* = 0.002). Vitamin C insufficiency is frequent in individuals undergoing home parenteral nutrition; therefore, there is a need to monitor plasma vitamin C levels, especially in patients on HPN with an inflammatory syndrome and only on CMVP.

## 1. Introduction

Home parenteral nutrition (HPN) is a lifesaving therapy for patients who cannot meet nutritional requirement intakes using the oral and/or enteral route. Owing to the absence of micronutrients in parenteral formulations, provision for vitamins and trace elements is required [1].

Vitamin C (ascorbic acid) is an essential water-soluble vitamin because, unlike many animals, humans are unable to synthetize it [2]. A regular and adequate intake of vitamin C is required owing to the low storage capacity of the human body. Vitamin C has several physiologic roles in humans such as being a cofactor for various enzymes including those involved in the metabolism of cholesterol to bile acid, carnitine biosynthesis, and hydroxylation of cortisol [3]. Scurvy is actually a clinical diagnosis of vitamin C deficiency, confirmed by low plasma vitamin C concentrations but yet not clearly define. It seems that physical signs of scurvy may only appear when plasma vitamin C decrease under 3 to 5 µmol/L, but there is not a definitive low vitamin C plasma level at which scurvy develops [2]. The French Health Authority (HAS) also defines a biological scurvy when the plasma vitamin C level is under 11 µmol/L [4].

Since vitamin C insufficiency could cause non-specific clinical symptoms such as anorexia, fatigue, muscle weakness, and arthralgia [5], that are quite common in HPN patients, clinicians cannot directly attribute these symptoms to vitamin C deficiency in this particular population.

The current guidelines for provision of vitamin C and monitoring in patients on long-term HPN are conflicting. Indeed, the European Society of Nutrition and Metabolism (ESPEN) recommends a standard dose of vitamin C of 100 mg to 125 mg, which is available in commercial multivitamin supplements [1]. The American guidelines recommend a dose of 200 mg of vitamin C per day [6], and the Australian guidelines 110 to 150 mg [7]. However, recommendations are scarce vis-à-vis the monitoring of vitamin C. For the Australian Society for Parenteral and Enteral Nutrition (AUSPEN), routine monitoring of vitamin C is not required when patients receive a parenteral multivitamin preparation [7], but there are no specific European or American recommendations [1,2].

In our HPN center, we usually monitor vitamin C in patients on long-term HPN and adjust parenteral intake when it is needed.

The aim of this study was to evaluate the vitamin C status of patients on long-term HPN for more than 6 months and to determine if an available European commercial multivitamin preparation (CMVP) with 125 mg of vitamin C (Cernevit^®^ Baxter S.A.) is sufficient to maintain the normal vitamin C status in patients on stable HPN.

## 2. Materials and Methods

### 2.1. Study Population

This study was conducted in the HPN referral center of Lyon, France. In our center, every patient who is on very long-term HPN is routinely hospitalized for nutritional assessment. All patients with HPN who were hospitalized for nutritional assessment from September 2007 to January 2020 were screened. Clinically stable adult patients receiving HPN or an intravenous fluid infusion at least 2 times per week for at least 6 months were retrospectively included. In case of multiple nutritional assessments, only the last one was considered for analysis.

### 2.2. Home Parenteral Nutrition

In our center, patients on HPN are on a commercial or on a personal compounded parenteral nutrition (PN). The preparations of personal PN admixtures are made by the hospital pharmacy or a manufacturer (Baxter Façonnage, Baxter S.A., Fretin, France) and stored in multilayered bags at 4 °C for up to seven days before administration. The manufacturer produces personal PN every seven days, so we can adjust the formulations of PN according to biological or clinical changes before their production begins. Vitamins and trace elements are added during the preparation of the admixtures for individual PN. For commercial PN, a multivitamin solution is injected into the PN bag shortly before administration.

### 2.3. Data Collection

Demographic and clinical variables were collected from electronic patient records, including age, gender, smoking history, body mass index (BMI), HPN causes (failure of enteral nutrition, short bowel syndrome (SBS), radiation enteritis, motility disorder, extensive mucosal disease), SBS causes, parenteral nutrition duration, number of weekly parenteral or intravenous fluid infusions, and parenteral vitamin C supplementation.

Whole blood samples were collected in an ongoing procedure at 7 a.m. after a night without PN and only with infusion of the same volume of fluid and electrolyte without vitamin and trace elements. Plasma vitamin C concentration was measured by high-performance liquid chromatography coupled to electrochemical detection with an H-Class Waters chromatographic system (Waters, St Quentin en Yvelines, Guyancourt, France) and Empower3_HF1_Enterprise software (version 7.30.00.00, Waters) [8]. To guarantee the stability of vitamin C, whole blood collected in lithium heparin tubes was centrifuged for a maximum of 3 h following blood collection. The supernatant was stabilized in a solution containing 6% of sulfosalicilyc acid to induce the precipitation of proteins, ethylenediaminetetraacetic acid (EDTA) to chelate bivalent ions which could disturb the electrochemical measurement, and N-ethylmaleimide to stabilize vitamin C in its reduced form. After acidification and stabilization, the diluted sample was frozen at a temperature below −18 °C and analyzed the following week. It was demonstrated, in a previous study, that vitamin C concentration was stable in acidified and stabilized plasma conserved below −18 °C for one month [9].

The normal range of vitamin C concentrations was 25 to 85 µmol/L. The normal minimal concentration of vitamin C was decreased from 30 µmol/L to 25 µmol/L during the study period (in 2008). We therefore chose 25 µmol/L to define a low level of vitamin C status. In addition, C-reactive protein (CRP), whose normal range is below 5 mg/L, was measured. Glomerular filtration rate (GFR) was estimated using the Chronic Kidney Disease Epidemiology (CKD-EPI) Creatinine Equation, and patients were classified according to their stage of chronic kidney disease.

### 2.4. Ethical Statement

The study was conducted according to the current French reference methodology, MR-003, and registered at the University Hospital Research Unit of Lyon, France.

### 2.5. Statistical Analyses

Quantitative variables are presented as mean ± standard deviation (sd) when normally distributed or as median [minimum–maximum value] when non-normally distributed. Qualitative variables are presented as number (percentage). The Pearson’s Chi-square (X^2^) test was used to compare categorical variables and the Student’s *t*-test or Welch’s *t*-test when two samples had unequal variances and were used for quantitative variables. All univariate predictors for a low vitamin C level with *p* < 0.2 were entered in a stepwise multivariate logistic regression model. The odds ratio (OR) and 95% confidence intervals (CI) are presented in the univariate and multivariate models. A *p*-value < 0.05 was considered significant. All analyses were performed using the Statistical Package for the Social Sciences (SPSS, IBM, Armonk, NY, USA), Windows software (version 19.0, Microsoft Windows, Chicago, IL, USA).

## 3. Results

### 3.1. Patients Characteristics

We screened all patients (*n* = 191) stably on HPN or an intravenous fluid infusion at least two times per week for at least 6 months between September 2007 and January 2020. Of these 191 patients, 186 met the inclusion criteria and were enrolled in the study.

The baseline characteristics of the 186 patients are shown in Table 1. Overall, the mean age was 56.9 ± 18.4 years. Seventy-nine patients (42.5%) were men and 36 (19.4%) were current smokers. The most common cause of HPN or intravenous fluid infusion was SBS (*n* = 13,673.1%). The principal cause of SBS was mesenteric ischemia (*n* = 6044.1%). The mean length of residual small bowel was 75.2 ± 52.4 cm.

Patients had a mean of 5.1 ± 1.8 intravenous infusions per week and a home parenteral support median of 33 (6–370) months. Eighty-five patients (45.7%) received compounded PN.

Concerning comorbidities, patients had neither hemochromatosis nor end-stage kidney disease.

The median parenteral intake of vitamin C was 125 mg (125–4000), and the median vitamin C level was 35 (5–84) µmol/L. One hundred and eleven (59.7%) patients received only the CMVP. Concerning an inflammatory syndrome, the median CRP level was 2.3 (0.1–77.5) mg/L, and one-third of the patients presented a CRP level above 5 mg/L.

### 3.2. Plasma Vitamin C Concentration

Among the 186 patients, 54 patients (29%) had plasma vitamin C concentrations below the normal minimal range, i.e., 25 µmol/L (5 to 24 µmol/L). None of our patients had a plasma vitamin C level under 5 µmol/L, and only 15 patients (8.1%) had a plasma vitamin C concentration below 11 µmol/L (definition of biological scurvy). As shown in Table 2, there were no significant differences regarding age, gender, indication for HPN, causes of SBS, SBS anatomy or length of residual small bowel, and number of parenteral infusions between patients with normal levels of vitamin C and those with low levels.

There were 79.6% (*n* = 43) of the patients with vitamin C insufficiency who received only the CMVP (*p* < 0.0001). Patients with a low level of vitamin C had a significantly lower duration of follow-up in our referral center (*p* < 0.0001) and a higher CRP concentration (*p* = 0.004) than patients with normal levels of vitamin C. As expected, average parenteral vitamin C supplementation was significantly higher in the group of patients with normal levels of vitamin C (*p* < 0.0001) than in the group with low levels. Patients with a normal range of vitamin C received a mean of 284.6 ± 230.1 mg (125 to 1400 mg) of vitamin C per parenteral infusion.

In multivariate analysis, only CRP (*p* = 0.001) and intravenous (IV) supplementation of only 125 mg of vitamin C through the CMVP (*p* < 0.0001) were independently associated with a low level of plasma vitamin C, as shown in Table 3.

### 3.3. Comparison of Patients Receiving Only the Commercial Multivitamin Preparation as Intravenous Vitamin C Supplementation

One hundred and eleven patients (59.7%) received only the CMVP as vitamin C supplementation (i.e., 125 mg of vitamin C). Patients with low plasma vitamin C concentrations and receiving only the CMVP were adminstered more personal compounded formulations of PN (*p* = 0.008) and had higher CRP levels (*p* = 0.002) than patients with a normal range of plasma vitamin C (Table 4). A borderline significant trend was observed for current smokers (*p* = 0.089) and patient with short bowel syndrome (*p* = 0.059).

## 4. Discussion

To date, the recommendations for vitamin C supplementation in HPN have been conflicting. In the literature, studies have reported vitamin C insufficiency in patients on prolonged total parenteral nutrition [10], and the same applied, as expected, to patients with SBS on intermittent parenteral nutrition [11]. To our knowledge, this study is the first to assess the vitamin C status in a large cohort of clinically stable adult patients on long-term HPN or intravenous fluid infusion.

We found that almost 30% of adult patients with HPN or intravenous fluid infusion had vitamin C insufficiency, whatever their intake; however, no one had had vitamin C deficiency with scurvy manifestation. These data were probably underestimated in our cohort. Indeed, in our referral center, patients on long-term HPN had plasma vitamin C concentration monitoring annually or every two years, and their nutrient intakes were adjusted accordingly when needed, which probably explains the significant difference in patients with low levels of vitamin C depending on the duration of follow-up in our center.

Our data showed an inverse relationship between CRP and plasma vitamin C. Elevation of CRP, even mild, reflects a low-grade systemic inflammation and increased oxidative stress. Vitamin C, by acting as an antioxidant, is well known to be lower in numerous environmental and health conditions. We hypothesized, like Massarenti et al. [12], that the low level of vitamin C in our population could reflect a depletion of vitamin C owing to its antioxidant and anti-inflammatory activities.

Surprisingly, there were no significative differences in vitamin C concentrations between patients with or without cancer, despite the fact that vitamin C deficiency occurs more frequently in patients with cancer [13,14,15] as a result of an increase of both oxidative stress [16] and inflammation [17]. These results can be explained by the low number of patients with active cancer on oncologic treatment (*n* = 6 patients, 3.2%). Smoking status, usually associated with vitamin C deficiency since it increases the turnover of ascorbic acid, [18,19,20], failed to reach significance in the subgroup receiving 125 mg of vitamin C (*p* = 0.089) probably for the same reasons or because of the adjustment in the intravenous intake of vitamin C after the previous screening.

In the subgroup of patients receiving 125 mg of vitamin C (only CMVP), individualized formulations of PN were significantly associated with vitamin C insufficiency (*p* = 0.008). This can be explained as follows: first, ascorbic acid in compounded parenteral nutrition is more degraded owing to the mixture of vitamins and trace elements [21], particularly in monolayered ethylvinyl acetate (EVA) bags [22,23] because of their permeability to oxygen. In our center, compounded PN bags are made using multilayered bags and are stored at 4 °C. Under these conditions, Dupertuis et al. showed that the half-life of ascorbic acid was 68.8 h [24]. In our hospital, Raja et al. (unpublished data) evaluated the content of vitamin C in multilayered bags with or without a UV filter. After 15 days of storage at 4 °C, 15% of the vitamin C content was lost. These data are different from those of Dupertuis et al. probably because Raja et al. used small PN bags containing only 1000 mL (vs. 2500 mL), which induced de facto a lesser area-to-volume ratio, which could decrease oxygen transmission through the bag wall. Secondly, as a result of the restricted access to individualized PN in our center, we selected the most severe patients for the administration of this kind of PN. Most of these patients had an ultra SBS requiring a high volume of PN, and their oral intake of vitamin C was negligible owing to the severity of malabsorption. In the same way, patients with vitamin C insufficiency who received the commercial polyvitamin preparation were those with SBS and without reach statistical significance (*p* = 0.059). It is probable that patients with SBS need higher doses of vitamin C because of their insignificant intestinal absorption. Moreover, the relationship between vitamin C insufficiency and higher inflammatory status without clinical explanation could be due to the existence of a low-grade inflammation induced by small intestine bacterial overgrowth [25,26]. An animal model showed that short bowel resection induced an early change in the microbiota in the remnant bowel and in the colon [27,28]. In a piglet model of short bowel syndrome, dysbiosis was associated with colonic inflammation [28], and Schall et al. demonstrated an upregulation of genes involved in cell proliferation, acute phase response signaling, immunity, and production of nitric oxide and reactive oxygen species in a zebrafish model of SBS [29]. In humans with SBS, Joly et al. found an alteration in the microbiota, with a high prevalence of *Lactobacillus* and a depletion of Clostridia and Bacteroidetes [30] and they showed that gut remodeling after bowel resection altered microbiota metabolism [31]. Low-grade inflammation could partly be the reflection of dysbiosis but could also be due to a pre-existing disease such as inflammatory bowel disease or cancer. Nevertheless, we could not eliminate a selection bias of our referral HPN center which enrolls more severe SBS patients who require more vitamin C.

As in any study, our study has limitations that should be mentioned. Firstly, it is limited by its retrospective and single-center design. Secondly, during the broad observation period, a change in vitamin C reference minimal range was introduced in 2008, decreasing the minimal concentration from 30 to 25 µmol/L. We therefore preferred to choose 25 µmol/L as the minimal value to ensure the validity of the study but with the probable risk of decreasing the number of patients with vitamin C insufficiency.

In the population who received 125 mg of vitamin C per bag (only CMVP), 38.7% had plasma vitamin C levels below the normal range. However, patients with normal plasma vitamin C concentrations received a mean of 280 mg of vitamin C per bag, which is more than usually recommended [1,6,7], suggesting that the amount of vitamin C contained in Cernevit^®^ is not sufficient to maintain vitamin C in the normal range during long-term HPN. However, we did not define a suitable dose of vitamin C because the study was not designed for that. There is a need for an interventional trial to evaluate the monitoring-directed supplementation of vitamin C and define the suitable dose of vitamin C and its effects on clinical symptoms in patients on long-term HPN. Nevertheless, our study suggests that patients on long-term HPN or intravenous fluid require a higher dose of vitamin C supplementation than currently recommended in Europe. The commercial polyvitamin preparation administered in our center failed to cover vitamin C requirements in one-third of the cases, in particular in patients with SBS and individualized PN.

## 5. Conclusions

This study revealed a high rate of vitamin C insufficiency in patients on long-term parenteral nutrition (29%). A commercial multivitamin preparation available in France failed to maintain the normal vitamin C status in more than one-third of the cases. These results therefore suggest that we must monitor plasma vitamin C level in patients on long-term HPN or intravenous fluid, especially in those with an inflammatory syndrome, even if it is mild.

## Figures and Tables

**Table 1 nutrients-12-01667-t001:** Patient characteristics.

Patient Characteristics	*n* = 186
**Demographic and biological characteristics**	
Male, *n* (%)	79 (42.5)
Age (years), mean ± sd	56.9 ± 18.4
BMI (kg/m^2^), mean ± sd	21.6 ± 4.4
Duration of home parenteral nutrition (months), median (min–max)	33 (6–370)
Duration of follow-up in the referral center (months), median (min–max)	27 (0–366)
Current smokers, *n* (%)	36 (19.4)
Number of parenteral infusions (per week), mean ± sd	5.1 ± 1.8
Compounded parenteral nutrition, n (%)	85 (45.7)
Parenteral vitamin C supplementation (mg/bag), median (min–max)	125 (125–1400)
Plasma vitamin C concentration (µmol/L), mean ± sd	37.7 ± 20.8
Patients only on CMVP, *n* (%)	111 (59.7)
C-reactive protein (mg/L), median (min–max)	2.3 (0.1–77.5)
**Indication for HPN, *n* (%)**	
Short bowel syndrome	136 (73.1)
Extensive parenchymal disease	16 (8.6)
Motility disorder	14 (7.5)
Postoperative malabsorption	10 (5.4)
Radiation enteritis	7 (3.8)
Enteral intolerance	3 (1.6)
**Short bowel syndrome causes, *n* (%)**	
Mesenteric ischemia	60 (44.1)
Surgical complications	31 (22.8)
Radiation enteritis	18 (13.3)
Crohn’s disease	9 (6.6)
Volvulus	9 (6.6)
CIPO	9 (6.6)
**Short bowel anatomy subgroup, *n* (%)**	
End jejunostomy	74 (54.4)
Jejunocolic anastomosis with no ileocecal valve	48 (35.3)
Jejunoileal anastomosis with ileocecal valve	14 (10.3)
Residual small bowel length (cm), mean ± sd	75.2 ± 52.4
**Cancer as comorbidity, *n* (%)**	
Without cancer	106 (57.0)
Cancer > 5 years	35 (18.8)
Cancer < 5 years without treatment	39 (21.0)
Cancer < 5 years on treatment	6 (3.2)
**Glomerular filtration rate, *n* (%)**	
GFR > 90 mL/min/1.73 m^2^	105 (56.5)
GFR 60–89 mL/min/1.73 m^2^	39 (21.0)
GFR 30–59 mL/min/1.73 m^2^	32 (17.2)
GFR 15–29 mL/min/1.73 m^2^	10 (5.4)

sd: standard deviation; BMI: Body mass index; CMVP: Commercial multivitamin preparation; HPN: Home parenteral nutrition; CIPO: Chronic intestinal pseudo obstruction; GFR: Glomerular filtration rate.

**Table 2 nutrients-12-01667-t002:** Patient characteristics according to their plasma vitamin C concentrations.

Patient Characteristics	Vit C Concentration		*p*
	Normal	Low	
	*n* = 132	*n* = 54	
**Demographic and biological characteristics**			
Male, *n* (%)	57 (43.2)	22 (40.7)	0.760
Age (years), mean ± sd	57.5 ± 18.5	55.6 ± 18.1	0.529
BMI (kg/m^2^), mean ± sd	21.5 ± 4.2	21.9 ± 4.9	0.527
Duration of home parenteral nutrition (months), mean ± sd	78.2 ± 88.0	54.4 ± 64.8	0.044
Duration of follow-up in referral center (months), mean ± sd	70.5 ± 79.6	38.0 ± 43.4	<0.0001
Current Smokers, *n* (%)	22 (16.7)	14 (25.9)	0.198
Number of parenteral infusions (per week), mean ± sd	5.1 ± 1.8	5.2 ± 1.8	0.676
Compounded parenteral nutrition, *n* (%)	66 (50.0)	35 (64.8)	0.066
Parenteral vitamin C supplementation (mg/bag), mean ± sd	284.6 ± 230.1	170.4 ± 109.8	<0.0001
Vitamin C plasma concentration (µmol/L), mean ± sd	47.1 ± 17.0	14.5 ± 5.6	<0.0001
Patients only on CMVP, *n* (%)	68 (51.5)	43 (79.6)	<0.0001
C-reactive protein (mg/L), mean ± sd	6.2 ± 11.6	14.2 ± 18.7	0.004
**Indication for HPN, *n* (%)**			
Short bowel syndrome	94 (71.2)	42 (77.8)	0.757
Extensive parenchymal disease	11 (8.3)	5 (9.3)	
Motility disorder	10 (7.6)	4 (7.4)	
Postoperative malabsorption	8 (6.1)	2 (3.7)	
Radiation enteritis	6 (4.5)	1 (1.8)	
Enteral intolerance	3 (2.3)	0	
**Short bowel syndrome causes, *n* (%)**			
Mesenteric ischemia	41 (43.6)	19 (45.2)	0.729
Surgical complications	24 (25.6)	7 (16.7)	
Radiation enteritis	11 (11.7)	7 (16.7)	
Crohn’s disease	7 (7.4)	2 (4.8)	
Volvulus	5 (5.3)	4 (9.5)	
CIPO	6 (6.4)	3 (7.1)	
**Short bowel anatomy subgroup, *n* (%)**			
End jejunostomy	51 (54.2)	23 (54.8)	0.896
Jejunocolic anastomosis with no ileocecal valve	34 (36.2)	14 (33.3)	
Jejunoileal anastomosis with ileocecal valve	9 (9.6)	5 (11.9)	
Residual Small Bowel Length (cm), mean ± sd	77.7 ± 52.3	69.5 ± 53.0	0.413
**Cancer as comorbidity, *n* (%)**			
Without cancer	76 (57.6)	30 (55.6)	0.843
Cancer >5 years	25 (18.9)	10 (18.5)	
Cancer <5 years without treatment	26 (19.7)	13 (24.1)	
Cancer <5 years on treatment	5 (3.8)	1 (1.8)	
**Glomerular filtration rate, *n* (%)**			
GFR > 90 mL/min/1.73 m^2^	72 (54.5)	33 (61.1)	0.573
GFR 60–89 mL/min/1.73 m^2^	27 (20.5)	12 (22.2)	
GFR 30–59 mL/min/1.73 m^2^	26 (19.7)	6 (11.1)	
GFR 15–29 mL/min/1.73 m^2^	7 (5.3)	3 (5.6)	

**Table 3 nutrients-12-01667-t003:** Univariate and multivariate models with a low plasma vitamin C concentration as the dependent variable and predictor variables as the independent variables.

Variable	Univariate Analysis			Multivariate Analysis		
	Odds Ratio	95% CI	*p*-Value	Odds Ratio	95% CI	*p*-Value
C-reactive protein ≥5 mg/L	3.092	(1.605; 5.957)	0.001	3.625	(1.801; 7.294)	<0.0001
CMVP	3.679	(1.747; 7.750)	0.001	4.292	(1.961; 9.393)	<0.0001
Duration of follow-up in the referral center (continuous variable)	1.009	(1.002; 1.015)	0.009			
Current smoker	1.750	(0.817; 3.748)	0.150			

**Table 4 nutrients-12-01667-t004:** Comparison of patients receiving only the commercial multivitamin preparation for their intravenous vitamin C supplementation according to their blood level of vitamin C.

Patient Characteristics	Vitamin C Concentration		*p*
	Normal	Low	
	*n* = 68	*n* = 43	
**Demographic and biological characteristics**			
Male, *n* (%)	23 (33.8)	19 (44.2)	0.273
Age (years), mean ± sd	61.9 ± 16.9	55.7 ± 18.5	0.070
BMI (kg/m^2^), mean ± sd	21.8 ± 4.7	22.3 ± 5.2	0.609
Duration of home parenteral nutrition (months), mean ± sd	40.8 ± 54.9	50.0 ± 67.8	0.435
Duration of follow-up in the referral center (months), mean ± sd	36.8 ± 38.8	31.4 ± 37.4	0.479
Current smokers, *n* (%)	10 (14.7)	12 (27.9)	0.089
Number of parenteral infusions (per week), mean ± sd	4.9 ± 1.8	5.3 ± 1.8	0.214
Compounded parenteral nutrition, *n* (%)	5 (7.4)	11 (25.6)	0.008
Plasma vitamin C concentration (µmol/L), mean ± sd	42.5 ± 15.0	14.2 ± 5.5	<0.0001
C-reactive protein ( mg/L), mean ± sd	4.7 ± 7.5	13.7 ± 17.0	0.002
**Indication for HPN, *n* (%)**			
Short bowel syndrome	44 (64.7)	35 (81.4)	0.059
**Short bowel anatomy subgroup, *n* (%)**			
End jejunostomy	24 (54.5)	19 (54.3)	0.983
Jejunocolic anastomosis with no ileocecal valve	15 (34.1)	12 (34.3)	
Jejunoileal anastomosis with ileocecal valve	5 (11.4)	4 (11.4)	
Residual small bowel length (cm), mean ± sd	95.8 ± 54.6	74.7 ± 54.9	0.099

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
