# Peer review of "Vitamin C in Home Parenteral Nutrition: A Need for Monitoring"

_nutrients, 2020, doi:10.3390/nu12061667_

Round 1

Reviewer 1 Report

This is an interesting study that reports an important nutrient deficiency in patients.  However, I have a few questions regarding the study design and data interpretation.

  1. Were Vit C concentration measurements conducted at the same time of day and same time either prior to infusion or post-infusion?  This could have an effect on Vit C concentration and samples collected at different times may induce confounding data.  Also, were Vit C concentrations measured at the same time i.e. all retrospective samples measured at the same time or were they measured in an ongoing procedure.  Vit C is easily degraded and so samples stored over a long period of time may well indicate a lower concentration than when collected.
  2. As mentioned, data comparison with other studies is hampered by differences in bag size, material and storage methods.
  3. The possible effects of enteric size (SBS) and bacterial colonies were only mentioned superficially.  These could potentially have a huge impact on plasma Vit C concentrations.  In addition, enteric bacteria are almost certainly involved in systemic inflammatory status and therefore affect CRP concentrations, independent of Vit C effect.
  4. As IV-administered Vit C has the potential to create a higher plasma concentration when compared to oral consumption, it would be prudent to screen for pre-existing conditions such as kidney disease or hemochromatosis that could be adversely affected by high Vit C levels.

Author Response

Dear  Reviewer,

Thank you for giving us the opportunity to revise our article, " Vitamin C in home parenteral nutrition: a need for monitoring to adjust supplementation !". We took great attention to the comments and requested modifications. Please find below answers to your comments.

We also acknowledge that all authors reviewed and approved the revised version of the article.

The manuscript, including related data and tables, has not been previously published and is not under consideration elsewhere.

Julienne Grillot, MD, on behalf of all the authors.

Response to Reviewer 1 Comments

Point 1. This is an interesting study that reports an important nutrient deficiency in patients.  However, I have a few questions regarding the study design and data interpretationWere Vit C concentration measurements conducted at the same time of day and same time either prior to infusion or post-infusion? This could have an effect on Vit C concentration and samples collected at different times may induce confounding data. Also, were Vit C concentrations measured at the same time i.e. all retrospective samples measured at the same time or were they measured in an ongoing procedure.  Vit C is easily degraded and so samples stored over a long period of time may well indicate a lower concentration than when collected

Response 1: Thank you for this relevant comment. We clarify this section in the methodology of the revised version:

Whole blood samples were collected in an ongoing procedure, at 7am, after a night without PN and only infusion of the same volume of fluid and electrolyte without vitamin and trace element.

Plasma vitamin C concentration was measured by high performance liquid chromatography coupled to electrochemical detection with H-Class Waters chromatographic system (Waters, St Quentin en Yvelines, France) and Empower3_HF1_Enterprise software (version 7.30.00.00, Waters) (7). To guarantee stability of vitamin C, whole blood collected on lithium heparin tube was centrifuged in a maximum of 3 hours following blood collection. Supernatant was stabilized in a solution containing 6% of sulfosalicilyc acid to induce precipitation of proteins, EDTA to chelate bivalent ions which could disturb electrochemical measurement and N-ethylmaleimide to stabilize vitamin C in its reduced form. After acidification and stabilization, diluted sample was frozen at <-18°C and analyzed the following week. It was demonstrated, in a previous study, that vitamin C concentration was stable in acidified and stabilized plasma conserved at <-18°C during one month (8).

Point 2. As mentioned, data comparison with other studies is hampered by differences in bag size, material and storage methods.

Response 2: We agree, and this explain why we unfortunately cannot compared our results with other studies.

Point 3. The possible effects of enteric size (SBS) and bacterial colonies were only mentioned superficially. These could potentially have a huge impact on plasma Vit C concentrations. In addition, enteric bacteria are almost certainly involved in systemic inflammatory status and therefore affect CRP concentrations, independent of Vit C effect.

Response 3: Thank you for your comment. We totally agree and developed this section in the revised discussion. As you can see below:

Animals model have shown that short bowel resection induce an early change in microbiota in the remnant bowel and also in the colon (27,28) . In a piglet model of short bowel syndrome, dysbiosis was associated with colonic inflammation (28) and Schall et al. demonstrated an upregulation in gene involved in cell proliferation, acute phase response signaling, immunity, and production of nitric oxide and reactive oxygen species in a model of SBS zebrafish (29). In SBS humans, Joly et al. found an alteration in microbiota with a high prevalence of Lactobacillus and depletion of Clostridia and Bacteroidetes (30) and they showed that gut remodelling after bowel resection alters microbiota metabolism (31). Low grade inflammation could partly be the reflect of dysbiosis but also be due to pre existing disease such as inflammatory bowel disease or cancer

Point 4. As IV-administered Vit C has the potential to create a higher plasma concentration when compared to oral consumption, it would be prudent to screen for pre-existing conditions such as kidney disease or hemochromatosis that could be adversely affected by high Vit C levels.

Response 4:Thank you for this comment, we actually screen kidney function using an estimation of the glomerular filtration rate (GFR) with CKD-EPI Creatinine Equation. Patients were also classified in stages of chronic kidney disease, as you can see in Table 1. In Table 2, we showed no significant difference concerning kidney function between patients with normal or low plasma vitamin C concentration.

In our study, median vitamin C level was 35 µmol/L with a maximum value of 84 µmol/L, still in the normal range of vitamin C concentration for our laboratory (25 to 85 µmol/L).

So we add the text in the Results section : “Concerning comorbidities that could be affected by high level of vitamin C, no patient had hemochromatosis nor end-stage kidney disease.”

We totally agree that there is a need to screen for condition like kidney disease or hemochromatosis that could be negatively affected by high level of vitamin C. This corroborate even more the need to monitor vitamin C, especially in those patients to adjust supplementation as close as possible.

Reviewer 2 Report

Thank you for the opportunity to review “Vitamin C deficiency in home PN”, report of an observational study evaluating vitamin C levels in patients receiving home PN or other IV fluids in home setting. Sorry for the delay due to COVID19 situations.

The prevalence and clinical importance of vitamin deficiencies in the  context of artificial nutrition and disease is a topic of high clinical relevance, yet not easy to study.

My major question is, does this observational study provide more information to clinicians and researchers regarding Vitamin C deficiency in HPN. To answer this question more information should be provided regarding:

- The clinical value of a measured blood level vitamin C with regard to true Vitamin C deficiency in chronically ill patients. The lower level of normal range as defined by clinical biologists may still be much higher than the level at which vitamin C deficiency becomes clinically relevant. Please provide more reference to published evidence validating measured blood levels of Vitamin C as a surrogate for Vitamin C deficiency. If I understand the results correctly, these blood Vitamin C levels may be rather a surrogate measure of inflammation? Also throughout the text replace “vitamin C deficiency” by “measured blood vitamin C levels below XXX”.

- Alternatively the authors might attempt to “validate” the applied serum Vitamin C methodology by relating the chosen cut-off with clinical signs of vitamin C deficiency in there patient population.

I haven’t got many other comments but the entire manuscript would benefit from critical editing to avoid potential confusion for the reader.

Example given:

  1. In the abstract “and use of CMVP were independently associated with a low plasma vitamin C concentration.” Reads as if commercial vitamin C preparations induce vitamin C deficiency. In the manuscript I understand that it is “use of CMVP only” so I shouldn’t stop using these supplements but combine them with others, right?
  2. Line 64: “In our center, patients on HPN are on commercial or compound parenteral nutrition (PN). The preparations of individual PN admixtures are made by the hospital pharmacy or a manufacturer (Baxter Façonnage, Baxter S.A.) and stored in multilayered bags at 4°C for up to seven days before administration. “ Does the manufacturer provides individual PN admixtures, on a patient by patients basis??
  3. In the abstract the authors state “duration of follow- up in our referral center (p = 0.009) and CMVP (p = 0.001) were associated with vitamin C deficiency.” An honest but alarming finding as if the longer they take care of patients, worse it becomes. But in the manuscript, I understand the opposite, it is, an inverse association between duration of follow up and vitamin C levels. A finding that again may suggest that early in illness inflammation is higher reflected by higher CRP and lower blood vitamin C levels
  4. In the abstract “Patients receiving only CMVP with a low plasma 25  vitamin C level significantly presented compounded HPN (p = 0.008) “ not clear how patients presented HPN?
  5. I wouldn’t use a “!” in titles of scientific manuscripts, particularly when the findings are even though very interesting still patterns in a retrospective observational study

Hopefully my comments will further improve this manuscript and the information gathered over many years by this expert nutrition team.

 Author Response

Response to reviewer 2 comments :

 Point 1: My major question is, does this observational study provide more information to clinicians and researchers regarding Vitamin C deficiency in HPN. To answer this question more information should be provided regarding:

- The clinical value of a measured blood level vitamin C with regard to true Vitamin C deficiency in chronically ill patients. The lower level of normal range as defined by clinical biologists may still be much higher than the level at which vitamin C deficiency becomes clinically relevant. Please provide more reference to published evidence validating measured blood levels of Vitamin C as a surrogate for Vitamin C deficiency. If I understand the results correctly, these blood Vitamin C levels may be rather a surrogate measure of inflammation? Also throughout the text replace “vitamin C deficiency” by “measured blood vitamin C levels below XXX”.

- Alternatively the authors might attempt to “validate” the applied serum Vitamin C methodology by relating the chosen cut-off with clinical signs of vitamin C deficiency in there patient population.

Response 1: Thank you for this major comment,

Firstly, in the revised version we clarify the definition of low plasma vitamin C and replace vitamin C deficiency by vitamin C insufficiency (i.e <25 µmol/L).

We developed the definition of insufficiency and scurvy in the revised manuscript

In the introduction section:

“Scurvy is actually a clinical diagnosis of vitamin C deficiency, confirmed by low plasma vitamin C concentrations but yet not clearly define. It seems that physical signs of scurvy may only appear when plasma vitamin C decrease under 3 to 5 µmol/L, but there is not a definitive low vitamin C plasma level at which scurvy develops (2). The French Health Authority (HAS) also defines a biological scurvy when the plasma vitamin C level is under 11 µmol/L (4).While Vitamin C insufficiency could cause non-specific clinical symptoms such as anorexia, fatigue, muscle weakness and arthralgia (5), quite common in HPN patients. That why clinicians could not only trust clinical signs of vitamin C deficiency in this particular population.

In the results section:

“In our cohort, none of our patients had a plasma vitamin C concentration under 5 µmol/L and only 15 patients (8.1%) were below 11 µmol/L.”

I haven’t got many other comments but the entire manuscript would benefit from critical editing to avoid potential confusion for the reader.

Example given:

Point 2: In the abstract “and use of CMVP were independently associated with a low plasma vitamin C concentration.” Reads as if commercial vitamin C preparations induce vitamin C deficiency. In the manuscript I understand that it is “use of CMVP only” so I shouldn’t stop using these supplements but combine them with others, right?

Response 2: We agree with your comment, and change CMVP by “intake of only 125mg of vitamin C” in the revised abstract to avoid confusion for the reader. We certainly do not want the readers to stop using CMVP, but warning them that in almost third of the case it isn’t sufficient to maintain vitamin C in the normal range. One may ask if vitamin C would be increase in European CMVP to 200mg like in the US. Unfortunately, we did not define a suitable dose of vitamin C because the study was not designed for that, as mentioned in the limits of our study.

We also added in the discussion :

“In the population who received 125mg of vitamin C per bag (only CMVP), 38.7% had plasma vitamin C levels below the normal range. While, patients with normal plasma vitamin C concentrations received a mean of 280mg of vitamin C per bag which is more than usually recommended (1,6,7) suggesting that the amount of vitamin C containing in Cernevit® is not sufficient to maintain vitamin C in the normal range on long term HPN.”

Point 3: Line 64: “In our center, patients on HPN are on commercial or compound parenteral nutrition (PN). The preparations of individual PN admixtures are made by the hospital pharmacy or a manufacturer (Baxter Façonnage, Baxter S.A.) and stored in multilayered bags at 4°C for up to seven days before administration. “ Does the manufacturer provides individual PN admixtures, on a patient by patients basis??

Response 3: The manufacturer produce personal PN every seven days, so we could adjust the formulation of PN according to biological or clinical change before the production begins.

In the revised version, we choose the term personal instead of individual.

Point 4:In the abstract the authors state “duration of follow- up in our referral center (p = 0.009) and CMVP (p = 0.001) were associated with vitamin C deficiency.” An honest but alarming finding as if the longer they take care of patients, worse it becomes. But in the manuscript, I understand the opposite, it is, an inverse association between duration of follow up and vitamin C levels. A finding that again may suggest that early in illness inflammation is higher reflected by higher CRP and lower blood vitamin C levels

Response 4: We totally agree that our formulation was not clear and change it in the revised version, as you can see below:

“In univariate analysis, C-reactive protein (p = 0.002) and intake of only 125mg of vitamin C (p = 0.001) were negatively associated with vitamin C level. Duration of follow-up in our referral center (p = 0.009) was positively associated with vitamin C level.”

Point 5: In the abstract “Patients receiving only CMVP with a low plasma 25  vitamin C level significantly presented compounded HPN (p = 0.008) “ not clear how patients presented HPN?

Response 5: All patients had HPN. Eighty-five patients (45.7%) had personal compounded HPN and one hundred and one patients (54.3%) had commercial PN.

We change compounded PN for “personal compounded PN” to ensure that no possible confusion could be created.

Point 6: I wouldn’t use a “!” in titles of scientific manuscripts, particularly when the findings are even though very interesting still patterns in a retrospective observational study

Response 6: Thank you for this relevant comment, the “!” was removed in the final title.

Round 2

Reviewer 2 Report

I would like to thank the authors for the consideration given to my comments!

Many issues are clarified now, including the definition of the "primary endpoint" vitamin C insufficiency. I agree that these results mandate more frequent monitoring, yet I don't know whether supplementation should be adapted in every patient with a low level detected or should be considered if also fatigue and other clinical signs are present. So based on the new information provided, I would remove "to adjust supplementation", likewise in the abstract and the conclusion. The urgent need for an interventional trial (randomized or cross over) evaluating fixed dose vitamin C supplementation versus monitoring-directed supplementation should be discussed. Such trial should focus on clinical endpoints combining fatigue and weakness.

All confusion has been solved. Some lines  need some additional English editing, examples.   

77 The manufacturer produce personal PN

133 patient had

Author Response

POINT 1: I would like to thank the authors for the consideration given to my comments!

Many issues are clarified now, including the definition of the "primary endpoint" vitamin C insufficiency. I agree that these results mandate more frequent monitoring, yet I don't know whether supplementation should be adapted in every patient with a low level detected or should be considered if also fatigue and other clinical signs are present. So based on the new information provided, I would remove "to adjust supplementation", likewise in the abstract and the conclusion. The urgent need for an interventional trial (randomized or cross over) evaluating fixed dose vitamin C supplementation versus monitoring-directed supplementation should be discussed. Such trial should focus on clinical endpoints combining fatigue and weakness.

RESPONSE 1: 

We agree and removed "to adjust supplementation" in the title, abstract and conclusion.

We also think that "There is a need for an interventional trial to evaluate monitoring-directed supplementation of vitamin C to define the suitable dose of vitamin C and the effect on clinical symptoms in patients on long-term HPN" and it was added to the discussion. 

POINT 2: All confusion has been solved. Some lines  need some additional English editing, examples.   

77 The manufacturer produce personal PN

133 patient had

RESPONSE 2: We would like to thank you for pointing out some errors.

The revised manuscript has been meticulously proofread to avoid spelling or grammar errors.